

# The tilt-effect in DOAS observations

Johannes Lampel[1,*], Yang Wang[1,2], Andreas Hilboll[3,4], Steffen Beirle[1], Holger Sihler[1,3], Janis Puķīte[1], Ulrich Platt[1,5], and Thomas Wagner[1]

[1]Max Planck Institute for Chemistry, Mainz, Germany
[2]Anhui Institute of Optics and Fine Mechanics, Chinese Academy Sciences, Hefei, China
[3]Institute of Environmental Physics (IUP), University of Bremen, Bremen, Germany
[4]Center for Marine Environmental Sciences (MARUM), University of Bremen, Bremen, Germany
[5]Institute of Environmental Physics, University of Heidelberg, Germany
[*]now at: Institute of Environmental Physics, University of Heidelberg, Germany

*Correspondence to:* J. Lampel (johannes.lampel@iup.uni-heidelberg.de), Yang Wang (y.wang@mpic.de)

**Abstract.**

Experience of differential atmospheric absorption spectroscopy (DOAS) shows that a spectral shift between measurement and reference spectrum is frequently required in order to achieve optimal fit results, while the straight forward calculation of the optical density proves inferior. The shift is often attributed to temporal instabilities of the instrument, but implicitly solved

the problem of the 'tilt-effect' discussed/explained in this paper.

Spectral positions of Fraunhofer and molecular absorption lines are systematically shifted for different measurement geometries due to an overall slope – or 'tilt' – of the intensity spectrum. The phenomenon has become known as 'tilt effect' for limb satellite observations, where it is corrected for in a first order approximation, whereas the remaining community is less aware of its cause and consequences.

It is caused by the measurement process because atmospheric absorption and convolution in the spectrometer do not commute. Highly resolved spectral structures in the spectrum first will be modified by absorption and scattering processes in the atmosphere. before they are recorded with a spectrometer, which convolves it with a specific instrument function. In the DOAS spectral evaluation process, however, the polynomial (or other function used for this purpose) accounting for broadband absorption is applied after the convolution is performed.

In this paper, we derive that changing the order of the two modifications of the spectra leads to different results. Assuming typical geometries for the observations of scattered sunlight and a spectral resolution of 0.6 nm, this effect can be interpreted as a spectral shift of up to 1.5 pm, which is confirmed in actual the analysis of the ground based measurements of scattered sunlight as well as in numerical radiative transfer simulations. If no spectral shift is allowed by the fitting routine, residual structures of up to $2.5 \times 10^{-3}$ peak-to-peak are observed. Thus, this effect needs to be considered for DOAS application aiming at RMS of

the residual of $10^{-3}$ and below.



# 1 Introduction

For a measured structured spectrum $s(\lambda)$ (e.g. scattered Sun light), the tilt effect emerges, because structures do not cancel out completely in the ratio of a measured spectrum $t(\lambda)$ relative to another spectrum with a different colour, which denotes the broad band spectral dependence. We show later that this can be interpreted as a spectral shift. This is due to the fact that the broad-band shape of the atmospheric transmission and the convolution with the instrument function do not commute (Wenig et al., 2005).

The tilt-effect was previously described and is explicitly corrected for by Sioris et al. (2003, 2004); Haley et al. (2004); Sioris et al. (2006); Krecl et al. (2006); McLinden and Haley (2008); McLinden et al. (2010); Rozanov et al. (2011) by including one additional tilt-effect pseudo absorber in the spectral analyses, whose magnitude was determined from the spectral fit. In their pioneering work, Sioris et al. (2003) named this effect 'tilt-effect' and corrected its impact by including a correction spectrum calculated using a radiative transfer model in the spectral analysis. However, Sioris et al. (2003) do not provide a mathematical derivation and rather estimates the effect's magnitude. It remains unclear if this effect is related to spectral undersampling and if it is also significant for other observations of scattered sunlight. Haley et al. (2004) also provide formulae for the correction spectrum, state that the effect is directly related to spectral undersampling and note that the effect is stronger for $NO_2$ retrievals in the blue spectral range than for $O_3$ in the green spectral range, due to smaller Fraunhofer lines. Rozanov et al. (2011) provide a mathematical derivation for the tilt effect correction spectrum (in Appendix B) and state that the tilt effect can be interpreted as a spectral shift. However, also here, only one fixed correction spectrum is used, which is scaled accordingly by the fitting routine.

We will derive here that the tilt-correction can be interpreted as a spectral shift for Gaussian instrument functions, which can however also vary with wavelength. We show that the calculated spectral shifts due to the tilt-effect agree with the observed shifts from DOAS analyses of ground-based measurements. As it turns out, the effect is not related directly to spectral undersampling (Chance et al., 2005) and not restricted to a certain wavelength range.

We observed, that for Multi Axis - Differential Optical Absorption Spectroscopy (MAX-DOAS) evaluations (e.g. Hönninger and Platt, 2002), which allow for a spectral shift of the measurement spectrum relative to a reference spectrum, systematic spectral shifts of up to 2 pm at a spectral resolution of the instrument of 0.6 nm are observed as shown in Figure 2 with the exact magnitude depending on the observation geometry. If no spectral shift is allowed in the fitting routine, residual structures of up to $2.5 \times 10^{-3}$ peak-to-peak are observed. Thus this effect needs to be considered for any DOAS application using a structured light source (such as the Sun) aiming at RMS of the residual of $10^{-3}$ and below. This is done implicitly in many DOAS retrieval codes by allowing for a spectral shift between measurement and reference spectrum. This option was originally introduced in the different analysis softwares to account for real shifts caused by instrumental instabilities (compare e.g. Peters et al., 2016). And in fact, the observed shifts derived from the spectral analysis were usually attributed to such instrumental stabilities only.

MAX-DOAS instruments typically contain thermally stabilized spectrometers, in order to avoid changes in their pixel to wavelength calibration. For such instruments, the spectral stability within one day has often a similar magnitude as the tilt-




effect (often less than a few pm). So called Fraunhofer reference spectra are recorded regularly at zenith viewing direction: these are used as reference for the spectral analysis. If Fraunhofer reference spectra are recorded each 10–15 minutes, then the change in spectral calibration of the instrument for the measurement spectrum relative to the Fraunhofer reference spectrum becomes small (typically $< 0.1$ pm, cf. Figure 2) and no larger spectral shift in the DOAS analysis can be explained by

instrumental instabilities any more. However, significant spectral shifts are still observed and are furthermore related to the telescope elevation angle of the MAX-DOAS observation. These can be explained in such cases by the tilt-effect as shown in subsection 3.4.

  When a measured spectrum is evaluated relative to another spectrum of the same setup, instrumental effects on the tilt are expected to cancel, as both spectra are influenced in the same way, e.g. by the efficiency of the grating and the detector.

However, if a measured spectrum is evaluated relative to a so-called Kurucz sun spectrum (as e.g. in Burton and Sawyer, 2016; Lübcke et al., 2016), the instrumentally induced tilt change can also lead to an apparent relative spectral shift.

  Another interesting aspect is that correction of the measured shifts including the tilt-effect will allow to estimate the spectral stability of passive DOAS instruments more precisely as shown in subsection 3.4.

  In section 2 we mathematical derive the expected spectral shift for a simplified instrument function. The expected spectral

shifts are compared in section 3 to field measurements. For these, as also in section 4 for the case of synthetic spectra, good agreement is found. Finally we discuss in section 5 different ways of how the tilt effect can be corrected. We provide examples and estimate its impact on the spectral retrieval.

# 2 Mathematical Derivation

## 2.1 Principle

A sketch of the principle of the tilt-effect is shown in Figure 1 where two individual $\delta$-shaped emission lines are used instead of a Sun spectrum.

  The two emission lines $\delta_1$ and $\delta_2$ are observed using a (virtual) spectrometer with a spectral resolution of 0.6 nm. This is shown as Gaussian peaks around each of the lines (dashed, blue and red). If both lines have the same intensity, the resulting total observed intensity (yellow) has its maximum in the middle of the two lines. If the lines are attenuated by a polynomial

$p(\lambda)$ (purple, in intensity space), the resulting total observed intensity (dark-grey) appears to be shifted in wavelength. This unrealistically steep broad band slope in $p(\lambda)$ was chosen to illustrate the effect, typically the slope of the polynomial in intensity space in DOAS observations is two orders of magnitudes smaller, as is the spectral shift due to the tilt-effect (see section 3).

## 2.2 Definitions

The instrument response function or instrument slit function $h(\lambda_0, \lambda)$ describes the response of the spectrometer for incoming radiation of wavelength $\lambda_0$ at the response wavelength $\lambda$ on the detector.





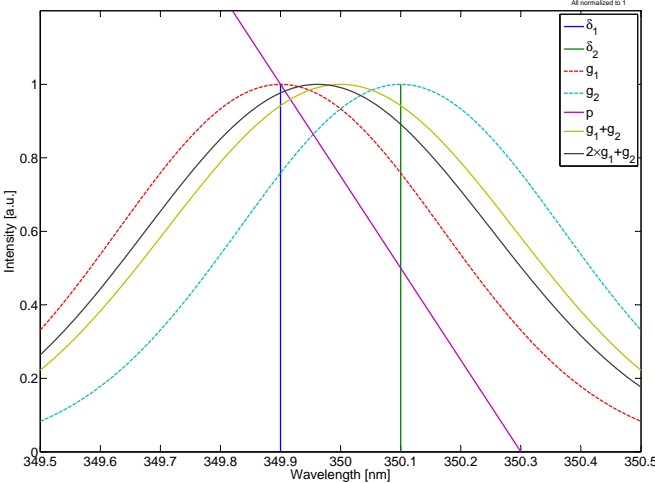

**Figure 1.** Illustration of the tilt-effect: an explanation is found in subsection 2.1. Note that all intensities except the polynomial are normalized to a maximum value of unity, for clarity.

Let $p(\lambda)$ be a polynomial in intensity space describing the broad-band change in the shape of the spectrum due to scattering processes and broadband absorption in the atmosphere.

Finally $k(\lambda)$ is a high resolution sun spectrum, e.g. from Chance and Kurucz (2010).

A low-resolution sun spectrum $s(\lambda)$ can be calculated from these quantities, where $\otimes$ denotes here the convolution operator.

5  For simplicity, $s(\lambda)$ is assumed to be direct sunlight with neither extinction nor absorption.

$$s(\lambda) = k(\lambda) \otimes h(\lambda_0, \lambda) = \int d\lambda_0 h(\lambda_0, \lambda) k(\lambda_0) \tag{1}$$

and with the wavelength dependent attenuation $p(\lambda)$, we obtain

$$t(\lambda) = (k(\lambda) p(\lambda)) \otimes h(\lambda_0, \lambda) = \int d\lambda_0 h(\lambda_0, \lambda) p(\lambda_0) k(\lambda_0) \tag{2}$$

The optical density which is typically fitted in DOAS applications (Platt and Stutz, 2008) is then

10  $$\tau(\lambda) = \frac{t(\lambda)}{s(\lambda)} \tag{3}$$

The tilt-effect as described in current literature describes the fact, that absorption structures in $s(\lambda)$ and $t(\lambda)$ (Fraunhofer lines and atmospheric absorbers on Earth) do not cancel completely when calculating the optical depth $\tau$, even if $p(\lambda)$ is smooth. We will show that it produces an apparent shift $\Delta\lambda$ of $t(\lambda)$ with respect to $s(\lambda)$. It is caused by the broad-band spectral



variation $p(\lambda)$, typically approximated by a polynomial in optical density, which does not commute with the convolution with the instrument function: $p(\lambda)s(\lambda) - t(\lambda) \neq 0$. In the next subsection, we therefore want to show the following equation:

$$p(\lambda)s(\lambda) \stackrel{!}{\approx} t(\lambda - \Delta\lambda) = t(\lambda) - \Delta\lambda \frac{\partial t(\lambda)}{\partial \lambda} + \mathcal{O}(\Delta\lambda^2) \tag{4}$$

Apart from the shift $\Delta\lambda$, higher orders $\mathcal{O}(\Delta\lambda^2)$ are neglected here. Note that the right-hand side of Equation 4 is similar to the 'tilt' definition in (Haley et al., 2004, equation 20), even though it is there not directly connected to a spectral shift.

## 2.3 Derivation

Without restriction of generality, a Gaussian instrument function is used in the following, as it has some useful analytical properties.

$$g(\lambda_0, \lambda) = \frac{1}{\sigma\sqrt{2\pi}} e^{-\frac{(\lambda-\lambda_0)^2}{2\sigma^2}} \tag{5}$$

Note that any instrument function can be represented by a sum of Gaussian functions (subsection 5.4) and that many instrument function are indeed close to Gaussian shape as e.g. in Beirle et al. (2016). To show a useful relation which is needed later, we set for simplicity $\lambda_0 = 0$ and we use a first order polynomial

$$q(\lambda) = 1 - w\lambda \tag{6}$$

We use $\frac{\partial g(\lambda)}{\partial \lambda} = -\frac{\lambda}{\sigma^2} g(\lambda)$ to reformulate:

$$q(\lambda)g(\lambda) = g(\lambda) - w\lambda g(\lambda) = g(\lambda) + w\sigma^2 \frac{\partial g(\lambda)}{\partial \lambda} \tag{7}$$

which then is $g(\lambda + \Delta\lambda) + \mathcal{O}(\Delta\lambda^2)$ with $\Delta\lambda = w\sigma^2$. We find that the spectral shift $\Delta\lambda$ is indeed proportional to the product of the tilt $w$ of the spectrum and the square of the width of the instrument function. $\mathcal{O}(\Delta\lambda^2)$ represents second order effects. This can also change of the effective shape of the instrument function (see subsection 5.3).

The average $w$ from Equation 6 (or later $\frac{\partial}{\partial\lambda}d(\lambda)$ with the DOAS polynomial $d(\lambda)$ from Equation 13) was found within 0.025–0.01 nm$^{-1}$ averaged over the fit interval using a fixed Fraunhofer reference spectrum for the MADCAT campaign described in section 3.

For measurements, $p(\lambda)$ is normally not linear in $\lambda$ due to the characteristics of Mie and Rayleigh scattering. Therefore the derivative $\frac{\partial}{\partial\lambda}p(\lambda)$ is not constant and the spectral shift $\Delta\lambda$ also depends on the wavelength $\lambda$ itself.

With Equation 7 we can calculate $t(\lambda)$ from Equation 2:

$$t(\lambda) = (k(\lambda)p(\lambda)) \otimes g(\lambda_0, \lambda) = \int d\lambda_0 g(\lambda_0, \lambda)p(\lambda_0)k(\lambda_0) \tag{8}$$



Taylor expansion of $p(\lambda_0)$ around $\lambda$ yields:

$$t(\lambda) = \int d\lambda_0 g(\lambda_0,\lambda)[p(\lambda) + (\lambda_0 - \lambda)\frac{\partial}{\partial\lambda'}p(\lambda')|_{\lambda'=\lambda} + \mathcal{O}(\lambda_0 - \lambda)^2]k(\lambda_0) \tag{9}$$

which is, with the shift from Equation 7 at wavelength $\lambda$ and neglecting higher order terms:

$$\approx p(\lambda)\int d\lambda_0 g(\lambda_0, \lambda + \Delta\lambda(\lambda))k(\lambda_0) = p(\lambda)s(\lambda + \Delta\lambda(\lambda)) \tag{10}$$

with

$$\Delta\lambda(\lambda) = \sigma^2\frac{1}{p(\lambda)}\frac{\partial}{\partial\lambda}p(\lambda) = \sigma^2\frac{\partial}{\partial\lambda}\ln(p(\lambda)) \tag{11}$$

Using that $p(\lambda)$ is defined in intensity space and is related to the DOAS polynomial $d(\lambda)$ in optical density space (logarithm of intensity) via

$$p(\lambda) = e^{-d(\lambda)} \tag{12}$$

we get

$$\Delta\lambda(\lambda) = -\sigma^2\frac{\partial}{\partial\lambda}d(\lambda) \tag{13}$$

This is the more general case of Equation 7 for non-linear DOAS polynomials.

## 2.4 Relation to the colour index

The colour index $CI(\lambda_a,\lambda_b)$ is defined by the ratio of intensities $I_a$ and $I_b$ at two distinct wavelengths $\lambda_a$ and $\lambda_b$ (see e.g. Sarkissian et al., 1991) and can be used to describe the 'tilt' of a spectrum in first order approximation.

$$CI(\lambda_a, \lambda_b) = \frac{I_a}{I_b} \tag{14}$$

Instead of analysing the DOAS polynomial, it is often sufficient to look at the difference in colour indices of measurement and reference spectrum, or in other words: at the 'tilt' of the spectrum (or part of a spectrum) as in Sioris et al. (2003). For a measurement spectrum $I'$ with a DOAS polynomial $d(\lambda)$ relative to the reference spectrum $I$, we get

$$CI'(\lambda_a, \lambda_b) = \frac{I_a e^{-d(\lambda_a)}}{I_b e^{-d(\lambda_b)}} \tag{15}$$





and thus we obtain using Equation 11

$$
\begin{aligned}
\text{CI' } - \text{CI} &= \frac{I_a\big(e^{-d(\lambda_a)} - e^{-d(\lambda_b)}\big)}{I_b e^{-d(\lambda_b)}} \\
&= \frac{I_a}{I_b}\frac{\lambda_a - \lambda_b}{e^{-d(\lambda_b)}}\frac{\partial}{\partial \lambda}e^{-d(\lambda)}|_{\lambda=\lambda_c}; \lambda_c \in [\lambda_a, \lambda_b] \\
&\approx -\frac{I_a(\lambda_a - \lambda_b)}{I_b}\frac{\partial}{\partial \lambda}d(\lambda)|_{\lambda=\lambda_c}; \lambda_c \in [\lambda_a, \lambda_b]
\end{aligned}
\tag{16}
$$

This means that the difference of colour indices between different spectra is proportional to the derivative of the DOAS polynomial at a certain point $\lambda_c$ within the fit interval. The derivative of the DOAS polynomial is again proportional to the apparent spectral shift due to the tilt-effect (Equation 13).

## 3 Measurements

For a spectral resolution of $\approx 0.6\,\text{nm}$ and typical DOAS polynomials, shifts of up to around 1 pm are expected using Equation 13. In this section, we will set the expected spectral shift due to the tilt-effect in relation to the spectral shift of the measurement spectrum in DOAS fits.

### 3.1 Measurement site

The Multi Axis DOAS - Comparison campaign for Aerosols and Trace gases (MAD-CAT) in Mainz/Germany took place on the roof of the Max Planck Institute for Chemistry (MPIC) during June and July 2013 [1]. The measurement site is located in the outskirts of Mainz and close to Frankfurt as well as several smaller towns. 11 research groups participated with the MAX-DOAS instruments. The intercomparison's aim primarily at the spectral retrieval of nitrogen dioxide ($NO_2$), formaldehyde (HCHO), nitrous acid (HONO) and glyoxal (CHOCHO), their azimuthal distributions and the retrieval of their respective vertical concentration profiles. Data from this campaign has been already published e.g. in Ortega et al. (2014), Lampel et al. (2015), Peters et al. (2016) and Wang et al. (2017).

### 3.2 Instrument description

We apply data obtained by an EnviMeS MAX-DOAS instrument during the MAD-CAT campaign. It is based on two Avantes ultra-low stray-light AvaSpec-ULS2048x64 spectrometers ($f = 75\,\text{mm}$) using a back-thinned Hamamatsu S11071-1106 detector. The spectrometer is temperature stabilized at $20.00\,^\circ C$ with deviations of $\Delta T < 0.02\,^\circ\text{C}$ at the temperature sensor. The UV spectrometer covered a spectral range of 294–458 nm at a FWHM spectral resolution of $\approx 0.6\,\text{nm}$ or $\approx 7$ pixels. The spectral stability was determined from the position of the Ca lines at around 393 and 397 nm and was typically better than $\pm 2\,\text{pm}$ per day and better than $\pm 5\,\text{pm}$ for the duration of the measurements from 6 June 2013 until 31 July 2013.

---

[1] http://joseba.mpch-mainz.mpg.de/mad_cat.htm



**Figure 2.** Measured shift (top panel) and colour index (340 nm, 370 nm) (bottom panel) as a function of time and observation elevation (colour coded) for one day (June 16th, 2013) during the MAD-CAT campaign relative to a Fraunhofer reference spectrum recorded close to local noon (thin blue vertical line). The thick blue line in the upper panel represents the pure instrumental shift after the shift introduced by the tilt-effect was removed (see text)





Mercury discharge lamp spectra recorded at different spectrometer temperatures yield a shift of the spectral calibration of this spectrometer type of about 4.5 pm/K. The maximum deviation of the spectrometer temperature from the nominal temperature was $\Delta T < 0.02\,\mathrm{K}$, thus less than 0.1 pm spectral shift can be attributed to temperature instability close to thermal equilibrium under ideal conditions.

During laboratory test measurements, a change of the spectral calibration of the instrument over time was found to be proportional to the temperature difference outside the thermally insulated spectrometer box and the spectrometer temperature and is therefore attributed to the residual temperature differences due to heat flux from the Peltier element through the spectrometer and the thermal insulation. We assume that this is the main reason for the variation of the inferred instrumental spectral shift in Figure 2.

Mercury discharge lamp spectra to obtain the instrument slit function $H(\lambda_0, \lambda)$ were recorded manually. No significant change of the instrument slit function shape was observed during the campaign.

The 1-D-telescope unit measures its elevation angle constantly using a MEMS acceleration sensor to determine the true vertical direction, and corrects the elevation angle, when it deviates from the nominal elevation angle. It has a vertical/horizontal field of view (FOV) of 0.2/0.8°. During daylight, spectra were recorded for 1 min each at 11 elevation angles of 90° (zenith), 30,

15, 10, 8, 6–1° respectively, as long as solar zenith angles (SZA) were smaller than 87°. Until a SZA of 100° zenith sky spectra were recorded at 90° telescope elevation. The exposure time was adjusted within the DOASIS (Kraus, 2006) measurement script to obtain spectra at a typical saturation of 50 %.

## 3.3 Analysis

Even though the tilt-effect is a general effect and not restricted to a certain wavelength range, we adapted here the HONO

retrieval settings suggested by Wang et al. (2017) for the spectral analysis. Similar results were obtained in other wavelength intervals (e.g. a glyoxal retrieval window from 432 to 458 nm).

The analysis of measured and synthetic spectra (see section 4) was done using the DOASIS software using a noon Fraunhofer reference spectrum. The literature cross-sections were convolved using the measured instrument slit function at 334 nm.

### 3.3.1 Shift and Squeeze parameters

The shift $a$ and squeeze $b$ (also called stretch) allow the DOAS fit to shift and squeeze cross-sections in order to minimize the fit RMS and e.g. compensate for instrumental instabilities and other factors which can influence the spectral calibration of the instrument.

This is parametrized typically in the following way:

$$\Delta\lambda_{shift}(\lambda) = a + b(\lambda - \lambda_0) + c(\lambda - \lambda_0)^2 + \mathcal{O}((\lambda - \lambda_0)^3) eq : shiftdef \tag{17}$$

Higher orders such as $c$ are often not used and set to zero.





| | T | $S_0$ | | HONO | | |
|---|---|---|---|---|---|---|
| Wavelength interval [nm] | Start | | | 335 | | |
| | End | | | 373 | | |
| $H_2O$ vapour | 298K | | | $\times$ | (*) | Lampel et al. (2017); Polyansky et al. (2017) |
| $O_4$ | 293K | | | $\times$ | | Thalman and Volkamer (2013) |
| $O_3$ | 223K | $1 \times 10^{18}$ molec cm$^{-2}$ | | $\times$ | | Serdyuchenko et al. (2014) |
| | 243K | $1 \times 10^{18}$ molec cm$^{-2}$ | | $\times$ | | |
| HCHO | | | | $\times$ | | Meller and Moortgat (2000) |
| HONO | | | | $\times$ | | Stutz et al. (2000) |
| BrO | | | | $\times$ | | Fleischmann (2004) |
| $NO_2$ | 293K | $1 \times 10^{16}$ molec cm$^{-2}$ | | $\times$ | | Vandaele et al. (1998) |
| | | | | $\times$ | | Linear and square terms according to Puķīte et al. (2010) |
| | 220K | $1 \times 10^{16}$ molec cm$^{-2}$ | | $\times$ | | |
| Ring Spectrum at | 273K | | | $\times$ | | DOASIS (Kraus, 2006) |
| | 243K | | | $\times$ | | based on Bussemer (1993) |
| Ring Spectrum $\cdot \lambda^4$ | | | | $\times$ | | Wagner et al. (2009) |
| Polynomial degree | | | | 5 | | |
| Add. Polynomial degree | | | | 1 | | |

**Table 1.** Retrieval wavelength intervals and reference spectra for the MAX-DOAS. $S_0$ denotes the SCD used for the $I_0$-correction during convolution, if applicable. (*) Water vapour absorption around 363 nm was not considered for the calculation and analysis of synthetic spectra.

The choice of $\lambda_0$ depends on the implementation, it is the minimum wavelength of the fit interval in DOASIS and the middle of the fit interval in the QDOAS software package (Danckaert et al., 2012). Chosing $\lambda_0$ in the middle of the fit range has the advantage that the corresponding base functions for shift and squeeze are linearly independent, which can be favourable in terms of numerical stability.

## 3.4 Results

The variation of the observed spectral shifts of the measurement spectrum e.g. during June 16th, 2013 is less than 4 pm, as can be seen from Figure 2. This translates to a spectral shift of less than $0.3\,\mathrm{pm\,h}^{-1}$ or less than $0.06\,\mathrm{pm}$ per elevation angle sequence. This accuracy allows to distinguish the shift due to the tilt-effect (up to 2 pm) within each elevation angle sequence



| Case | 1 | 2 | 3 | 4 | 5 | 6 |
|------|---|---|---|---|---|---|
| Free shift parameters | shift, squeeze | shift | none | corr. spectrum | corr. spectrum and shift | none |
| Tilt-effect correction spectrum | | | | × | × | × |
| Shift [pm] | $1.1 \pm 0.1$ | $0.99 \pm 0.04$ | (0) | (0) | $0.019 \pm 0.04$ | (0) |
| Squeeze | $1.00 \pm 3.4 \times 10^{-6}$ | (1) | (1) | (1) | (1) | (1) |
| RMS $[10^{-4}]$ | 2.83 | 2.84 | 4.30 | 2.79 | 2.79 | 2.82 |
| dSCD HONO $[10^{14}$ molec cm$^{-2}]$ | 2.64 | 2.48 | 0.18 | 1.99 | 2.01 | 2.03 |
| $\sigma_{\mathrm{fit}}$ HONO $[10^{14}$ molec cm$^{-2}]$ | 2.46 | 2.47 | 3.67 | 2.39 | 2.44 | 2.46 |

**Table 2.** DOAS fit results from June 16th, 2013 at 04:46 UTC for a spectrum at 2° elevation for different settings of the spectral shift of the reference spectrum and with and without a tilt-effect correction spectrum (subsection 5.2). To minimize photon noise, four subsequent elevation angle sequences were co-added. Values in round brackets denote fixed values for shift and squeeze. Fit residuals and a fit of the correction spectrum are shown in Figure 3. The average shift within the fit interval due to the tilt-effect calculated from the DOAS polynomial itself amounts to 1.14 pm and a squeeze of $1 + 8 \times 10^{-7}$, which could not be resolved from the measurement data. (squeeze de6inition from DOASIS, $\lambda_0 = \lambda_{min}$, see Equation 17)

from instrumental instabilities. The resulting correlation of measured shift and calculated shift determined from the DOAS polynomial is shown for the complete dataset (June and July 2013) in Figure 4 evaluated relative to the next zenith Fraunhofer reference spectrum. The shift due to the tilt-effect was calculated from the DOAS polynomial using Equation 13. As the shift varies with wavelength, we used here the average shift calculated on an equidistant grid of 0.1 nm within the fit interval.

5     A correlation coefficient $R^2 = 0.83$ and slope of $0.95 \pm 0.02$ was observed. The y-axis intercept of the fitted polynomial was small and amounted to 0.05 pm, which is less than 1/1000 of the spectral width of a detector pixel. The small deviation of the fitted slope of the correlation from unity can result from a slightly varying instrument function width within the fit interval ($< 2\%$) and effective weighting of the shift at different wavelengths due to variable depth of the Fraunhofer lines ($< 3\%$). The average measurement error of the shift (estimated by twice the fit error following Stutz and Platt (1996)) amounts to 0.03 pm

10  ($< 1.5\%$). Furthermore the instrument function of the spectrometer used here is not exactly Gaussian.

    Having shown that the shifts are mostly caused by the tilt-effect, this allows to correct the measured shift of the reference spectrum for the shift by the tilt-effect to obtain the instrumental shift at higher precision, also during unsupervised field measurements and without the need for calibration lamps. This is also shown in Figure 2. The resulting instrumental shift is stable until about 9:00 UTC (with a standard deviation of less than 0.1 pm), a time after which the room temperature changed,

15  as probably the door was opened and the temperature outside the instrument started changing. The gap in measurement data







**Figure 3.** Corresponding plots of fit residuals to the cases 1,3 and 5 from Table 2 and the tilt-effect correction spectrum for case 5.

around noon is caused by a restart of the measurement routine. As also the temperature stabilization routine was restarted, the gap is followed by a shift in the spectral calibration of 0.4 pm, as the temperature control needed a few minutes to stabilize. This effect would not have been as clearly visible without correction for the tilt-effect.

For an individual spectrum recorded at an elevation angle of $2°$ the fit results are shown in Table 2 using a reference spectrum recorded in the same elevation angle sequence at an elevation angle of $90°$. Here five cases are distinguished with different numbers of free parameters for shift, squeeze and the explicitly calculated tilt-effect correction spectrum according to Equation 19 using the DOAS polynomial obtained from a fit without considering the tilt-effect.



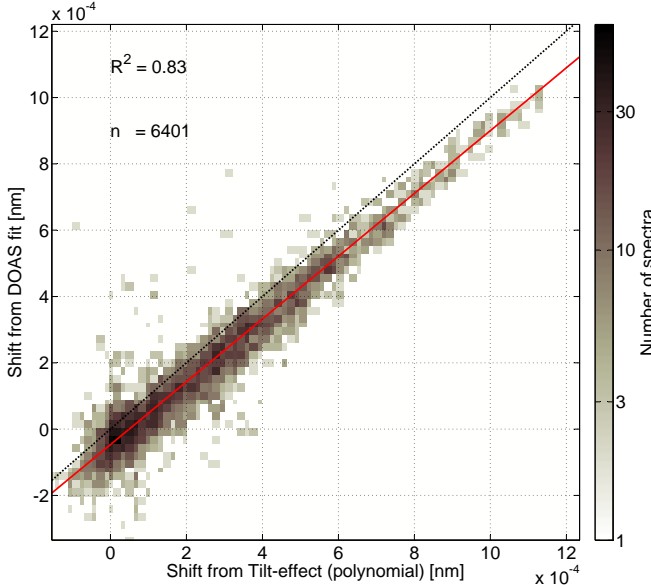

**Figure 4.** Correlation of the shift determined from the DOAS polynomial according to the tilt-effect and measured relative shift of the measurement spectrum to the following zenith sky spectrum (in order to minimize the influence of instrumental instabilities). To reduce the scatter of the data points further, four subsequent elevation angle sequences were co-added.

## 4 The 'tilt-Effect' in synthetic Spectra

Additionally the tilt-effect is demonstrated for synthetic spectra, in order to exclude any instrumental influences.

### 4.1 Calculation of synthetic spectra

All simulations were conducted with the radiative transfer model SCIATRAN Rozanov et al. (2014), version 3.6.0 (03 Dec

5  2015). SCIATRAN was operated in `raman` mode to simulate intensities of scattered sunlight in Mainz, Germany (49.99°N, 8.23°E)), including the effect of rotational Raman scattering in the Earth's atmosphere. The scalar radiative transfer problem was solved in a pseudo-spherical atmosphere (i.e., the solar beam is treated in spherical geometry, while the scattered or reflected beam is treated in plane-parallel geometry) using the discrete ordinate method. The simulations were conducted with 0.01nm spectral sampling, and Raman lines were calculated using the forward-adjoint approach and binned to the 0.01nm

10  wavelength grid (Rozanov and Vountas, 2014).

Absorptions by the trace gases ozone ($O_3$), nitrogen dioxide ($NO_2$), formaldehyde (HCHO), bromine oxide (BrO), nitrous acid (HONO), and by the $O_2$–$O_2$ collision complex ($O_4$) were considered; the respective cross-section references are the ones also used for analysis in Table 1. Aerosols were assumed to be mostly scattering, having an optical depth (AOD) of 0.135, an asymmetry factor of 0.68, and a single scattering albedo (SSA) of 0.94.





The simulated observation geometry was similar to the measurement sequences as described in subsection 3.2. A more detailed description, also of the concentration height profiles, can be found in Wang et al. (2017).

Water vapour absorption according to Lampel et al. (2017) and Polyansky et al. (2017) was not considered for the synthetic spectra, but was compensated in the measured data. A detailed analysis can be found in Wang et al. (2017).

## 4.2 Results

The spectral analysis was performed in analogy to subsection 3.3. The absorption of water vapour in the UV was not considered for the calculation of the spectra and thus also not in the spectral analysis.

The synthetic spectra represent measurements of an ideal instrument without any changes of the wavelength calibration due to external influences. Therefore the initial expectation of the analysis of the synthetic spectra was that no shift is needed in the spectral analysis between reference spectrum and measurement spectrum. However, as described in section 2, some spectral shift was found and needed to be compensated for.

Fits with a RMS of more than $4 \times 10^{-4}$ during twilight were filtered out, as saturation and radiative transfer effects of stratospheric ozone absorption increased the residuals of the fits significantly and have the potential to modify the calculated shift values. The correlation of calculated and fitted shift for the remaining 120 spectra due to the tilt-effect was very good with $R^2 = 0.9993$. Shifts of up to 1.2 pm due to the tilt-effect were found. The shift from the DOAS fit was about 2.1% larger than from the calculation of the tilt-effect. However, the average measurement error of the shift amounts to 0.02 pm and is thus of similar magnitude.

The small discrepancy could be also caused by the fact that the influence of rotational Raman scattering is calculated differently in DOASIS (according to Bussemer (1993); Platt and Stutz (2008) from the convolved, synthetic spectrum itself) and SCIATRAN (according to Rozanov and Vountas (2014) at the higher spectral resolution of 0.01 nm before convolution).

Overall the very good agreement of theoretically expected and calculated spectral shifts shows also the validity of the derivation of the tilt-effect.

## 5 Discussion - Correction of the tilt-effect

Even for a perfect MAX- or zenith sky DOAS instrument (as shown in section 4), the tilt-effect needs to be considered and corrected for. Typically it is corrected for by allowing a shift between measurement and reference spectrum. As the shift at each wavelength depends on the derivative of the broad band spectral dependence, which is usually corrected by the DOAS polynomial, an additional squeeze (and higher orders of the spectral shift) of the measurement spectrum can be necessary, depending on the desired magnitude of the residual. This is discussed in subsection 5.1.

The spectral shift depends on the spectral resolution of the instrument (see Equation 13), in fact it is proportional to the square to the spectral resolution.

Another approach is to calculate the effective shift spectrum from the explicit calculation of the commutator of polynomial and convolution, or in other words the difference between $p(\lambda)s(\lambda)$ and $t(\lambda)$. This approach is discussed in subsection 5.2.



## 5.1 Shift and squeeze

The apparent change in the wavelength determination due to the tilt-effect can be determined from the DOAS polynomial using Equation 13. The shift, squeeze and higher order parameters can then be determined by a polynomial fit of $\Delta\lambda(\lambda)$ using the QDOAS definition of $\lambda_0$ of squeeze and higher orders (see Equation 17). For each of the parameters of the polynomial

(corresponding to shift, squeeze, quadratic squeeze etc.), the maximum shift inside the fit range can be determined and can then be used for estimating the residual structure which is caused by the tilt-effect. This shift, converted to the corresponding optical depth, is shown in Figure 5. For typical applications (FWHM= $0.6\,\mathrm{nm}$, RMS$> 1 \times 10^{-4}$) it is therefore sufficient to allow shift and squeeze between measurement and reference spectrum in order to correct for this effect. The conversion factor $\alpha_{OD}$ from shift to optical density was determined from the pseudo-absorber of the spectral shift within the fit interval for the

given spectral resolution of the instrument and amounted in this case to $1.5\,\mathrm{nm}^{-1}$.

$$\alpha_{OD} = 2 \times \left| \frac{\partial s(\lambda)}{\partial \lambda} \frac{1}{s(\lambda)} \right|_{max} \tag{18}$$

## 5.2 Tilt-effect correction spectrum

For a known DOAS polynomial $d(\lambda)$, a correction spectrum $c(\lambda)$ can be calculated to compensate for the tilt-effect. This implies that an iterative fit process is performed and means thus higher computational costs. The correction spectrum $c(\lambda)$ is

the difference between two synthetic sun spectra calculated from a highly resolved solar atlas, one where the attenuation with the $p(\lambda) = e^{-d(\lambda)}$ in intensity space is applied before and one after the convolution operation:

$$c(\lambda) = p(\lambda)s(\lambda) - t(\lambda) \tag{19}$$

To use it in the DOAS fit as a pseudo-absorber (PA), it can be converted to optical density space by division with $s(\lambda)$

$$c_{PA}(\lambda) = \frac{c(\lambda)}{s(\lambda)} \tag{20}$$

This correction spectrum, introduced in the fit shown in Table 2, was indeed found in the spectral fit and reduced the shift of the reference spectrum from $1.1\pm0.1$ pm (case 2) to $0.019\pm0.04$ pm (case 5). As the calculation from Equation 19 also provides the absolute magnitude of the effect, this correction spectrum does not even need to be fitted as in previous publications, but can be applied directly (case 6). This can, if the instrument itself is stable, potentially reduce the degrees of freedom of the fit and thus result in lower detection limits. This could however not be observed here for measurement data.

The DOAS polynomial can be determined with sufficient precision without correcting the tilt-effect, as a small spectral shift can be represented via Taylor expansion by an individual spectrum, which is dominated by narrowband contributions (Beirle et al., 2013) as it is defined via the derivative with respect to wavelength of the respective spectrum. To test this, the



**Figure 5.** Peak-to-peak optical density caused by shift, squeeze and higher order squeeze due to the tilt-effect, using the dataset from Figure 2. The shift due to the tilt-effect was calculated from the DOAS polynomial for the corresponding mean wavelength of each pixel within the fit interval. Then a polynomial of third order was fitted to this data to calculate the corresponding shift, squeeze and higher order terms and thus the corresponding peak-to-peak ODs caused by the tilt-effect. It can be seen that shift and squeeze already compensate for most of the effect. The colour scale is the same as in Figure 2.





DOAS polynomial was determined for the spectrum shown in Table 2. This polynomial was used to calculate the correction spectrum. The absolute magnitude of the resulting DOAS polynomials differed relative to each other by up to 3%. Calculating the correction spectrum from the second DOAS polynomial results in a second correction spectrum which differs absolutely with an OD of $6 \times 10^{-5}$. Therefore further iterations of the fitting process are not needed in this case.

Note that this approach might need to consider also strong absorbers present in the observed spectra, which are not present in the solar atlas spectrum. This can play a role for ozone and sulfur dioxide absorption in the UV range and strong absorbers such as $H_2O$ and $O_2$ in the red and near-IR spectral range. A potential disadvantage is that this calculation requires knowledge of the exact instrument slit function, which is implicitly included in the first approach (subsection 5.1 and see also subsection 5.3). As often the spectral shift of the instrument also needs to be accounted for, shift and squeeze need to be implemented in any

case, which can make the calculation of explicit correction spectra in most cases obsolete. This choice depends on the desired precision of the result for very small RMS (compare Figure 5).

The correction spectra calculated using the DOAS polynomial include apart from the tilt-effect induced shift also the effect of the squeeze parameter and higher orders. Therefore a correction spectrum needs to be calculated corresponding to each fit. As seen from Figure 5 applying shift and squeeze is sufficient for most applications, but calculation of the correction spectrum

can reduce the impact of the tilt-effect even further, as seen in Table 2. Here the RMS of the default fit using shift and squeeze of the reference spectrum (1) is reduced by using an explicitly calculated correction spectrum slightly by 1 % (4), even though the number of degrees of freedom of the fit stayed constant (cases 1+5). When only the correction spectrum was used, and the shift fixed to zero, assuming no shift between Fraunhofer reference and measurement spectrum (cases 1+4), the RMS is the same, but the HONO fit error is reduced.

## 5.3   The influence of the instrument slit function

As shown in Equation 13 for the case of a Gaussian instrument response function, the spectral shift depends on the spectral resolution of the instrument, in fact it is proportional to the square to the spectral resolution. A real instrument function $h(\lambda_0, \lambda)$ is in general not a Gaussian function, but can be approximated by N Gaussian functions of different widths $\sigma_i$ shifted by $\Delta\lambda_{0i}$ and weighted by $w_i$, as it is typically also measured at finite spectral resolution.

$$h(\lambda_0, \lambda) = \sum_i^N w_i g_{\sigma_i}(\lambda_0 + \Delta\lambda_{0i}, \lambda) \qquad (21)$$

As summing and convolution are interchangeable, Equation 1 can be written as a sum over different $s_i(\lambda) = k(\lambda) \otimes g_{\sigma_i}(\lambda_0 + \Delta\lambda_{0i}, \lambda)$ using Equation 21. To these Gaussian instrument function the derivation of the tilt-effect applies individually. However, as the derivative with respect to $\lambda$ in the Taylor series for the spectral shift in Equation 4 also commutes with the sum, the shift calculated from $s(\lambda)$ also correctly compensates for the tilt-effect for non-Gaussian instrument functions.





### 5.4 Instrument slit function changes due to tilt-effect

As already pointed out for Equation 7, the squeeze and second order effects of the tilt-effect also lead to a slight modification of the effective instrument slit function's shape, apart from the spectral shift. Based on the DOAS polynomials obtained from the fits of measurements from June 16th (compare Figure 2) and using an initial Gaussian instrument slit function, the effective

instrument slit function was numerically calculated and fitted again with a Gaussian function. The first order tilt-effect shift was reproduced within $2 \times 10^{-8}$ nm. The relative width of the instrument function varied by up to $5 \times 10^{-4}\%$. For an absorber with a differential OD of unity, this results in an OD of less than $5 \times 10^{-6}$ and is therefore negligible.

### 5.5 Pixel-wavelength calibration of spectra

As the tilt-effect will influence all spectra recorded at low resolution, it will also have an effect on the spectral calibration of

scattered sunlight spectra, if done by fitting it to a high-resolution solar atlas, as e.g. Chance and Kurucz (2010). As will be shown in subsection 5.6, the effect on retrieved trace gases is typically negligible, as the expected shifts due to the tilt-effect are also here of the order of less than a few pm.

Note that also other calibration methods, as e.g. the calibration using line emission spectra, have uncertainties: If the position of the emission lines is determined by fitting Gaussian peaks, the fit error of the center of the peak also typically amounts to

2-3 pm, as the shape of the observed emission line is rarely Gaussian (e.g. Liu et al., 2015; Beirle et al., 2016). The width of a single pixel for the measurements shown above is typically 60 pm or larger. The variation between different mercury emission lamps is with 0.07 pm significantly smaller than the tilt-effect itself (Sansonetti and Reader, 2001).

The center-of-mass of an emission peak can be more accurately determined when the emission peaks are not undersampled.

### 5.6 The impact of the 'tilt-effect' on the spectral retrieval of trace-gases

The impact of the 'tilt-effect' on the spectral retrieval of trace gases is twofold: If the tilt-effect is not corrected for, the remaining residual structures can cause deviations for retrieved trace gases. The shift induced on the measurement spectrum is the same as for the absorbers, as similar considerations apply as well to the convolution of trace gases as to the convolution of the Fraunhofer spectrum. However, if the shift of the trace gases is not determined from the fit, but from a fit of the Fraunhofer reference spectrum to a solar atlas (typically with a different 'tilt' or colour indices), small shifts of the order of a few pm can

occur, which are not the same for the absorbers.

Using a pseudo-absorber for the spectral shift of $NO_2$ $\partial/\partial\lambda\sigma_{NO_2}(\lambda)$, we obtain a residual OD for a shift of 2 pm due to the tilt-effect of the $NO_2$ absorption cross-section of 0.2%. Thus 1.5% differential absorption by $NO_2$, which corresponds to a dSCD of $1 \times 10^{17}$ molec cm$^{-2}$ can result in a systematic residual structure due to the tilt-effect of $3 \times 10^{-4}$, which is often acceptable.





## 6 Conclusions

Based on a theoretical analysis as well as on measured and simulated scattered sunlight spectra we have shown that the tilt-effect can cause artificial shifts and and enhanced residuals, which are introduced by the the fact that any modification of the broadband spectral variation of a spectrum (e.g. caused by atmospheric scattering processes) does not commute with

the convolution with the instrument slit function. Thus an effective shift between measurement and reference spectra can be observed. This effect is called 'tilt-effect' according to Sioris et al. (2003). In the context of limb satellite observations, this effect was mathematically described by Rozanov et al. (2011). We showed that the spectral shift due to this effect is proportional to the square of the instrument resolution $\sigma$ and the slope of the broad band spectral shape, which can be described by the so called DOAS polynomial, which accounts for broad band spectral differences between the measured spectrum and

the Fraunhofer reference spectrum (e.g. caused by Mie and Rayleigh scattering and broadband absorptions). It is not directly connected to spectral undersampling as stated in previous publications.

A shift between measurement and reference spectrum is typically allowed for in DOAS retrievals and motivated by instrumental instabilities. We show that the shift caused by the tilt-effect is significantly larger than typical instrument shifts within one elevation sequence and that the main reason to allow for this shift is eventually the tilt-effect. For measured as well as for

simulated, synthetic spectra a good correlation between fitted and calculated shifts due to the tilt-effect is found.

For ground-based passive DOAS instruments with a spectral resolution of 0.6 nm and DOAS fits with a residual RMS of more than $10^{-4}$, we estimate that the tilt-effect can be compensated for by allowing for a shift and squeeze term. For DOAS fits with a residual RMS of less than $10^{-4}$, which can be obtained by co-adding a large number of spectra, higher order terms for the parametrization of the wavelength shift might be necessary. The shift due to the tilt-effect is typically not constant

with respect to wavelength $\lambda$ within the fit intervals, as it is proportional to the derivative of the so called DOAS polynomial (Equation 13). For observation geometries which show larger differences in colour indices, such as satellite limb observations, such corrections might even be necessary if the requirements on the magnitude of the residual are less strict.

Alternatively, using the known instrument function, correction spectra can be explicitly calculated for a given DOAS polynomial or approximated from a given the difference in colour indices between measurement and reference spectrum, similarly

as suggested in previous publications.

The effect is generally present for spectroscopic measurements at medium spectral resolution with wavelength dependent attenuation. Therefore the same effect can be expected for active measurements (e.g. Cavity-enhanced or Long-Path DOAS measurements) as well.

*Acknowledgements.* We thank the MAD-CAT-Team for support during the MAD-CAT campaign. We thank Klaus Pfeilsticker and Alexey

Rozanov for helpful comments during the preparation of the manuscript.



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
