# Peer review of "The tilt-effect in DOAS observations"

_Atmospheric Measurement Techniques, 2017_

## Referee Comment (RC1) · Anonymous Referee #1 · 4 Jul 2017

This paper is in good shape, but here are a few suggestions:

*Scientific comments*

Eqs.1-2: Why are the authors integrating over $\lambda_0$ rather than $\lambda$?

Eq. 18: Where does the factor of 2 come from? Is it because the optical depth is quantified peak-to-peak?

Table 2: Relative uncertainty on the dSCD is larger for case 4-6 than case 1. If the relative errors are improved by accounting for the tilt-effect when measuring glyoxal, this might be the better molecule for Table 2.

p2: I suggest a rewording of "not directly related to spectral undersampling". How about "not related to Doppler shifts"? I think spectral undersampling worsens the tilt-effect as spectrally wider pixels are equivalent to worsening the spectral resolution and this study already shows the dependence on spectral resolution. If the authors could test the effect of coarser spectral binning, this would improve the paper (relates to p2L14-15). Currently it appears as if the authors dispute the conclusion of Haley et al. of a relationship with between tilt-effect and spectral undersampling, based on p2L22. In any case, this statement at p2L22 ("As it turns out, …") should be in the conclusions section, not in the introduction.

Also there should be a caveat stated that the derivation has some limits: if the spectral resolution is so poor that the solar/telluric absorption features are no longer resolved, I would expect the tilt effect to be less important for such an extreme situation.

Conclusion: Do the authors think the tilt-effect could extend to lab spectroscopy where extinction is important (e.g. high pressure measurements, or in cases where there is some condensation/aerosol)?.

*Minor corrections*

p3L14: "In section 2 "-> "In section 2,"

p3L23: might be appropriate and more consistent with the start of the sentence to refer to $g_1$ and $g_2$ rather than "dashed, red and blue".

Eq. 5: Even though this is a well-known equation, $\sigma$ should be defined. Note that $\sigma_{fit}$ appears later (p11). This could be confusing.

p9, Eq. 17. I assume that "eq: shiftdef" is some kind of leftover from latex conversion? Otherwise this notation is strange to me.

p11: Table 2 caption: de6inition -> definition

dSCD should be defined as "differential slant column density" in this caption since this is the first appearance of the abbreviation. $\sigma_{fit}$ should be defined and maybe a symbol other than $\sigma$ should be used

(see earlier comment relating to Gaussian equation). The last two sentences of the current caption should be moved into the main body of the paper. I am confused whether 1.14 ppm is the shift for case 1 since an extra significant digit is provided. To which case does the squeeze value in the last sentence of the caption apply? It seems like it is case 1, but then the squeeze values are inconsistent ($10^{-6}$ vs $10^{-7}$). Most importantly, extra words are needed somewhere in the caption. What I have understood is that the first row below the header row indicates whether a tilt-effect calculated correction has been applied (i.e. cases where this correction has been applied are marked with an 'x'). If I have understood correctly, then it should be stated in the caption or in the text when Table 2 is first referenced (not at p15L22) that any case with an x has been corrected for the tilt-effect *before the DOAS fit*.

p12L5: This sentence mentions 5 cases "…where a DOAS fit is performed "without considering the tilt-effect". Yet, in Table 2, certainly cases 4-5 appear to include the tilt-effect in the fit.

P12L5: The presentation would be improved if the 19th equation in the current manuscript appeared before P12L5.

Section 4 title: tilt-Effect -> tilt-effect

p14L25: corrected for -> corrected

p15L20: fit shown -> fit results shown

p17L27: function -> functions

p18L17: is with 0.07 ppm -> is 0.07 ppm,

p19L15: simulated, synthetic -> simulated

Haley et al. ref: O3 and NO2 should contain subscripted numbers.

Sioris et al. (2006) ref: {NO2} -> $NO_2$

---

## Referee Comment (RC2) · Anonymous Referee #3 · 31 Jul 2017

**Review: The tilt-effect in DOAS observations**

*Johannes Lampel, Yang Wang, Andreas Hilboll, Steffen Beirle, Holger Sihler, Janis Putike, Ulrich Platt, Thomas Wagner*

**General comments**

The manuscript *The tilt-effect in DOAS observations* concerns with the effect of the non-commutativity of atmospheric absorption and convolution. The authors could show that this effect, labeled "tilt-effect", manifests itself as wavelength shift which was, up to now, solely attributed to instrumental shifts. This manuscript hence offers new insights in understanding and interpreting this error source when doing DOAS like data analysis. This well structured and fluently written manuscript is ideally suitable to be published in the journal *Atmospheric Measurement Techniques*. Only a few comments and minor suggestions need to addressed before publication.

**Specific comments**

1. **P2, L27**
   2.5e-3 peak-to-peak residual structure. Where is this coming from? Is it related to section 5.2?

2. **P4, Figure 1**
   Figure 1 in the manuscript has the goal to illustrate the tilt-effect. The good concept behind the figure is, however, not straight forward catchable (lines, colors, label size are not chosen adequately). I attached a re-design sketch, which could serve as a motivation.

3. **P9, L8**
   It would be clearer for the reader to mention at this point that the visualized instrumental spectral shift is rectified from the tilt effect already.

4. **P10, L8**
   ... the shift due to the tilt-effect (up to 2 pm)... Where is the 2 pm coming from? How relates this number to the estimation of $\approx$ 1pm based on equation (13), mentioned in P7, L7?

5. **P11, Table 2**
   From section 5.2 the difference between case 4 and 6 becomes clear, but not before. The reader would benefit from a more explicit description of the cases in the table and the caption.

6. **P11, L8**

[Figure]

Figure 1: Redesign proposal for Figure 1, P4.

How is the percentage impact of instrument function width and shift weighting estimated?

7. **P12, Figure 3 caption**
   The reader would benefit from a more detailed description to subfigure 4.

8. **P13, L13**
   At which wavelength are the aerosol parameters defined?

9. **P17, L1 to L4**
   If I have understood this correctly, the authors want to explain the iterative process to calculate the final tilt-correction spectrum, by first utilizing the tilt-uncorrected DOAS polynomial to get the first tilt-correction spectrum, and so on. I suggest to reformulate this paragraph to make the procedure more clear.

10. **P17, L9 to L19**
    The authors state that calculating the correction spectra is in most cases obsolete when shift and squeeze is implemented in the fitting anyway. Also the change in the RMS is discussed for the cases in Table 2. Providing that I have got the correct meaning of Table 2, cases 1, 2 (shift and squeeze applied) yield significantly different dSCD HONO values than cases 4, 5, 6 (with tilt-correction spectrum applied). Based on this the method used to correct for the tilt-effect (shift, squeeze or correction spectrum) seems to be rather important considering trace gas dSCD (usually the primary retrieval product). What is the authors comment about that?

11. **P18, section "The impact of the "tilt-effect" on the spectral retrieval of trace-gases"**
    This relates to comment 10. In my opinion the impact on the retrieved trace-gas amount is the most important aspect. Thus I think the manuscript would benefit if this section would be outlined with a little bit more detail, especially the way how the impact of the tilt-effect on trace gas retrievals is quantified.

12. **P19, section "Conclusions"**
    It would further round up the conclusions section if the authors conclusion about the impact of the tilt-effect on the accuracy of dSCD of trace gases would be added.

*Technical corrections*

1. **P3, L23**
   ... around each of the lines (dashed, **cyan** and red) ... (possibly obsolete, see comment 2. in former section)

2. **P8, Figure 2**

   When I am not mistaken, the bottom figure is never mentioned (or referred to) in the text explicitly. I suggest rectify this or to remove the bottom figure.

3. **P9, L10**

   Instrument slit function "H" should be lowercase "**h**" as introduced earlier in the text.

4. **P9, L20**

   Table 1 is not mentioned yet in the text. This could be a place to do this.

5. **P9, L29**

   Eq (17), LaTeX typesetting issue

6. **P11, Table 2 caption**

   Last sentence: squeeze definition

7. **P17, L27**

   To these Gaussian instrument functions the derivation ...

8. **P18, L28**

   ..., can result in a

9. **P19, L2**

   ... artificial shifts **and** enhanced residuals ...

10. **P19, L24**

    ... approximated from a given difference ...

---

## Referee Comment (RC3) · Anonymous Referee #2 · 7 Aug 2017

Lampel et al. present an important study showing the relationship between the so-called Tilt effect and the DOAS shift. It is an important work which is of interest for the DOAS community. The methodology is scientifically sound. I think this study should be published in AMT and I have only few minor comments:

- I find the paper could be improve in its structure. The author sometimes refer to figures or equations that are only properly introduced few pages later and I found it hard to follow.

- Eq.3 : the optical density commonly refers to the log-ratio of t and s. Please harmonize throughout the paper

- Eq. 17: there is an editing issue "eq:shiftdef"

- Table 1: For O3 cross-sections, I guess $S_o=1e19$ molec. cm-2 and not 1e18

[Figure]

molec.cm-2. It is not clear what is "Add. Polynomial degree".

- Table 2: typo: (squeeze de6inition-> squeeze definition)

---

## Author Comment (AC1) · 11 Oct 2017

**Answers to Reviews of Lampel et al, 'The tilt-effect in DOAS observations', AMTD 2017**

Johannes Lampel,Yang Wang,Andreas Hilboll,Steffen Beirle,Holger Sihler,Janis Puķīte,Ulrich Platt,Thomas Wagner

**1 Introduction and general comments**

**We would like to thank the editor and the three reviewers for their comments, suggestions and corrections. All comments together improved greatly the discussion manuscript. We considered each of the points, which are answered below together with a description of the related changes to the manuscript.**

(Numbers of equations, figures, lines and pages refer to the discussion manuscript, if not mentioned otherwise. **Authors' reponses are written in bold face**, the referees' text is shown in normal face.)

**2 Review #1**

This paper is in good shape, but here are a few suggestions:

**We thank the reviewer for the overall positive assessement.**

**2.1 Scientific comments**

1. Eqs.1-2: Why are the authors integrating over $\lambda_0$ rather than $\lambda$?

   **$\lambda$ is used on the left side of the equations, thus we integrate over $\lambda_0$. Which one of both $\lambda$s is used as the first argument of the instrument slit function $h(\lambda_0, \lambda)$ is then a matter of definition, we chose here $h(\lambda_0, \lambda)$.**

2. Eq. 18: Where does the factor of 2 come from? Is it because the optical depth is quantified peak-to-peak?

   **Yes, it is indeed because the proportionality factor was intended here for the relation between shift and peak-to-peak optical depth within the fitting interval. We added 'peak-to-peak' to the text before this equation.**

3. Table 2: Relative uncertainty on the dSCD is larger for case 4-6 than case 1. If the relative errors are improved by accounting for the tilt-effect when measuring glyoxal, this might be the better molecule for Table 2.

   **As the tilt-effect is a general effect under the given circumstances, we don't see any reason to chose another absorber, as this yields similar results in terms of the apparent spectral shift due to the tilt-effect. HONO was chosen as the related publication ([Wang et al., 2017]) is published and relies on the same synthetic data. The MAD-CAT glyoxal intercomparison is still in preparation by I. Ortega et al.**

4. p2: I suggest a rewording of not directly related to spectral undersampling. How about not related to Doppler shifts? I think spectral undersampling worsens the tilt-effect as spectrally wider pixels are equivalent to worsening the spectral resolution and this study already shows the dependence on spectral resolution. If the authors could test the effect of coarser spectral binning, this would improve the paper (relates to p2L14-15). Currently it appears as if the authors dispute the conclusion of Haley et al. of a relationship with between tilt-effect and spectral undersampling, based on p2L22. In any case, this statement at p2L22 (As it turns out, ) should be in the conclusions section, not in the introduction. Also there should be a caveat stated that the derivation has some limits: if the spectral resolution is so poor that the solar/telluric absorption features are no longer resolved, I would expect the tilt-effect to be less important for such an extreme situation.

**The reviewer is correct that both the spectral binning due to the pixel size as well as the representation of the finite spectral resolution of the instrument can be realized by a convolution operation. For both of these operations the considerations made here for the tilt-effect due to the finite spectral resolution of the instrument apply. However for practical purposes and typical current DOAS instruments which are not suffering from significant spectral undersampling, the tilt-effect due to spectral undersampling is a minor effect compared to the tilt-effect due to the finite spectral resolution of the instrument. It has to be however separated from the effects of spectral undersampling which has as pointed out in ([Chance et al., 2005]) other reasons than the tilt-effect itself.**

**We move the statement at p2L22 to the conclusions.**

**It should be noted that the undersampling correction as described in [Slijkhuis et al., 1999] could be rather named 'Linearization of the spectral shift between measurement and reference spectra for the special case of undersampled spectral data'. The spectral shift here was primarily motivated by Doppler shifts between Sun and earthshine spectra, but simultaneously also corrected (in parts) for the tilt-effect. We therefore added the following paragraph as a new subsection to the revised manuscript:**

**Previously the tilt-effect was also related to spectral undersampling ([Chance et al., 2005]): As e.g. described in [Slijkhuis et al., 1999] for spectral data from satellite, a spectral shift between the observed measurement and reference spectra was introduced in order to correct for Doppler-shifts between these. As these shifts are small compared to the spectral resolution of the instrument (typically $< 5$pm), the spectral shift can be linearized and directly calculated from a high-resolution sun spectrum (such as e.g. [Chance and Kurucz, 2010]) in order to include also artefacts of spectral undersampling. In [Slijkhuis et al., 1999] the DOAS fit finally contained this linearized shift as well as the non-linear shift and squeeze parameters of the measurement spectrum relative to the reference spectrum. Also the tilt-effect**

**introduces a spectral shift of similar magnitude, and is corrected (in first order approximation) in the same way. This potentially led to confusion in the available literature. The derivation of the tilt-effect shown above does however not depend at all on the properties of the spectral binning of the instrument and can therefore be considered independent of the undersampling effects.**

5. Conclusion: Do the authors think the tilt-effect could extend to lab spectroscopy where extinction is important (e.g. high pressure measurements, or in cases where there is some condensation/aerosol)?.

   **This could also play a role there. However, lab measurements are often done at higher spectral resolution, where the the apparent shift then significantly smaller (Eq. 13). We added a paragraph to the conclusions:**

   **It affects any medium resolution spectroscopic application where the spectral evaluation involves a step where the convolution and effects like scattering, which lead to a broad-band variation of the shape of the spectrum, are commuted. Laboratory measurements of trace gas absorptions are however often done at higher spectral resolution, which minimizes the apparent shift of the tilt-effect due to the relation shown in Eq. 13.**

**2.2 Minor corrections**

1. p3L14: "In section 2 "→ "In section 2,"

   **Changed**

2. p3L23: might be appropriate and more consistent with the start of the sentence to refer to g1 and g2 rather than "dashed, red and blue".

   **This point is now obsolete, as the figure has been redesigned as suggested by reviewer #2, see Figure 1.**

3. Eq. 5: Even though this is a well-known equation, $\sigma$ should be defined. Note that $\sigma_{fit}$ fit appears later (p11). This could be confusing.

   **$\sigma$ is now defined as property of the Gaussian function and $\sigma_{fit}$ is now explicitly mentioned in the caption of Table 2.**

4. p9, Eq. 17. I assume that eq: shiftdef is some kind of leftover from latex conversion? Otherwise this notation is strange to me.

   **Fixed.**

5. p11: Table 2 caption: de6inition → definition

   **Fixed.**

6. dSCD should be defined as differential slant column density in this caption since this is the first appearance of the abbreviation. $\sigma_{fit}$ should be defined and maybe a symbol other than $\sigma_{fit}$ should be used (see earlier comment relating to Gaussian equation). The last two sentences of the current caption should be moved into the main body of the paper. I am confused

whether 1.14 ppm is the shift for case 1 since an extra significant digit is provided. To which case does the squeeze value in the last sentence of the caption apply? It seems like it is case 1, but then the squeeze values are inconsistent (10-6 vs 10-7). Most importantly, extra words are needed somewhere in the caption. What I have understood is that the first row below the header row indicates whether a tilt-effect calculated correction has been applied (i.e. cases where this correction has been applied are marked with an x). If I have understood correctly, then it should be stated in the caption or in the text when Table 2 is first referenced (not at p15L22) that any case with an x has been corrected for the tilt-effect before the DOAS fit.

**We modified the table and extended the caption in order to clarify the meaning and the results shown in this table. The value 1.14 ppm is calculated from the DOAS polynomial as already stated in the caption. The table itself shows DOAS fit results. The squeeze values are consistent within the DOAS fit error. We added the meaning of the crosses ($\times$) to the caption of the table instead of p15l22.**

7. p12L5: This sentence mentions 5 cases where a DOAS fit is performed without considering the tilt-effect. Yet, in Table 2, certainly cases 4-5 appear to include the tilt-effect in the fit.

   **The DOAS polynomial itself was determined without considering the tilt-effect. This DOAS polynomial was then used to calculate the tilt-effect correction spectrum used in cases 4–6. p17l1-4 we stated that the effect of the tilt-effect on the DOAS polynomial itself is practically negligible. We split the sentence to make this more clear.**

8. P12L5: The presentation would be improved if the 19th equation in the current manuscript appeared before P12L5.

   **Actually Eq. 19 just follows from Eq. 4. on page 5.**

9. Section 4 title: tilt-Effect → tilt-effect

   **Fixed.**

10. p14L25: corrected for → corrected

    **Fixed.**

11. p15L20: fit shown → fit results shown

    **Fixed.**

12. p17L27: function → functions

    **Fixed.**

13. p18L17: is with 0.07 ppm → is 0.07 ppm,

    **Fixed.**

14. p19L15: simulated, synthetic → simulated

    **Fixed.**

15. [Haley et al., 2004] ref: O3 and NO2 should contain subscripted numbers.

    **Fixed.**

16. [Sioris et al., 2006] ref: NO2 → NO2

    **Fixed.**

**3 Review #2**

Lampel et al. present an important study showing the relationship between the socalled tilt-effect and the DOAS shift. It is an important work which is of interest for the DOAS community. The methodology is scientifically sound. I think this study should be published in AMT and I have only few minor comments:

**3.1 Minor comments**

1. I find the paper could be improve in its structure. The author sometimes refer to figures or equations that are only properly introduced few pages later and I found it hard to follow.

   **We restructured parts of the manuscripts slightly according to the suggestions by all three reviewers and especially improved the description of Table 2.**

2. Eq.3 : the optical density commonly refers to the log-ratio of t and s. Please harmonize throughout the paper

   **This basic is of course true and later used in a correct way (eq 12,18) and surprising that nobody saw that except you. This is now fixed here.**

3. Eq. 17: there is an editing issue eq:shiftdef

   **fixed.**

4. Table 1: For O3 cross-sections, I guess So=1e19 molec. cm-2 and not 1e18 molec.cm-2. It is not clear what is Add. Polynomial degree.

   **As the general settings were applied using a current reference Fraunhofer spectrum, typical ozone column densities are in the $10^{18}$ molec cm$^{-2}$ range (the dSCD for 40ppbv ozone along 10km close to the ground are of this order of magnitude). Add. Polynomial degree means additive polynomial degree and is now explicitly mentioned. We added 'e.g. [Peters et al., 2016]' as a reference, even though it is mentioned in various previous publications.**

5. Table 2: typo: (squeeze de6inition→ squeeze definition)

   **fixed.**

**4 Review #3**

**4.1 General comments**

The manuscript The tilt-effect in DOAS observations concerns with the effect of the non-commutativity of atmospheric absorption and convolution. The authors could show that this effect, labeled tilt-effect, manifests itself as wavelength shift which was, up to now, solely attributed to instrumental shifts. This manuscript hence offers new insights in understanding and interpreting this error source when doing DOAS like data analysis. This well structured and fluently written manuscript is ideally suitable to be published in the journal Atmospheric Measurement Techniques. Only a few comments and minor suggestions need to addressed before publication.

**4.2 Specific comments**

1. P2, L27 2.5e-3 peak-to-peak residual structure. Where is this coming from? Is it related to section 5.2?

   **The value is actually taken from Figure 5. The estimate of the magnitude of the tilt-effect calculated from the DOAS polynomial was based on typical DOAS polynomials and not extreme cases, which explains the difference between this number and the number given at the beginning of section 3.**

2. P4, Figure 1 Figure 1 in the manuscript has the goal to illustrate the tilt-effect. The good concept behind the figure is, however, not straight forward catchable (lines, colors, label size are not chosen adequately). I attached a re-design sketch, which could serve as a motivation.

   **Thank you for this nice plot, this really looks better. We applied your ideas and inserted a new and improved figure.**

3. P9, L8 It would be clearer for the reader to mention at this point that the visualized instrumental spectral shift is rectified from the tilt-effect already.

   **We added 'which was already corrected for the shift introduced by the tilt-effect' to this sentence.**

4. P10, L8 ... the shift due to the tilt-effect (up to 2 pm)... Where is the 2 pm coming from? How relates this number to the estimation of $\approx$ 1pm based on equation (13), mentioned in P7, L7?

   **see above**

5. P11, Table 2 From section 5.2 the difference between case 4 and 6 becomes clear, but not before. The reader would benefit from a more explicit description of the cases in the table and the caption.

   **We now explicitly mention the differences between the cases as indicated by row 2+3 of the table in its caption.**

6. P11, L8 review: the tilt-effect in doas observations How is the percentage impact of instrument function width and shift weighting estimated?

[Figure]

Figure 1: New version of Figure 1

**1. The variation of the instrument slit function within the fit interval was determined from the recorded mercury emission line spectra. 2. The effect of the weighting according to the depth of the Fraunhofer structures was estimated by calculating the weighted and the normal mean of the shift due to the tilt-effect within the fit interval. We added this as a short description prior to this estimate.**

7. P12, Figure 3 caption The reader would benefit from a more detailed description to subfigure 4.

   **We added 'The tilt-effect correction spectrum is shown in red, the sum of it and the residual is show in grey.'**

8. P13, L13 At which wavelength are the aerosol parameters defined?

   **We added 'All aerosol parameters were assumed to be constant over the whole wavelength range.'.**

9. P17, L1 to L4 If I have understood this correctly, the authors want to explain the iterative process to calculate the final tilt-correction spectrum, by first utilizing the tilt-uncorrected DOAS polynomial to get the first tilt-correction spectrum, and so on. I suggest to reformulate this paragraph to make the procedure more clear.

   **We only estimated the differences of DOAS polynomials from fits which do correct for the tilt-effect and do not correct for it in order to estimate the error made in the calculation. We also wrote this in the introducing sentence of this paragraph. We modified 'The absolute magnitude of the resulting DOAS polynomials differed relative to each other by up to 3%.' to 'The absolute magnitude of the resulting DOAS polynomials with and without**

**correcting for the tilt-effect differed relative to each other by up to 3%.'**

10. P17, L9 to L19 The authors state that calculating the correction spectra is in most cases obsolete when shift and squeeze is implemented in the fitting anyway. Also the change in the RMS is discussed for the cases in Table 2. Providing that I have got the correct meaning of Table 2, cases 1, 2 (shift and squeeze applied) yield significantly different dSCD HONO values than cases 4, 5, 6 (with tilt-correction spectrum applied). Based on this the method used to correct for the tilt-effect (shift, squeeze or correction spectrum) seems to be rather important considering trace gas dSCD (usually the primary retrieval product). What is the authors comment about that?

    **The changes of the observed dSCDs are clearly within the fit error for all cases in which the tilt-effect was corrected for. When not corrected for, the change in dSCD is of similar magnitude as the fit error.**

11. P18, section The impact of the tilt-effect on the spectral retrieval of trace-gases This relates to comment 10. In my opinion the impact on the re-trieved trace-gas amount is the most important aspect. Thus I think the manuscript would benefit if this section would be outlined with a little bit more detail, especially the way how the impact of the tilt-effect on trace gas retrievals is quantified.

    **We added the following text to this paragraph, which hopefully also answers this comment:**

    **For the case of HONO and a spectrum with a apparent shift due to the tilt-effect of 1 pm the results are shown in ??. It becomes clear that the overall effect of the tilt-effect on the retrieved HONO dSCDs is small and within the measurement error in this case, for this absorber and for this instrument. However, as the residual RMS and thus the fit error is significantly reduced, the correction of this effect is crucial for a correct determination of measurement errors and detection limits. If the shape of the structures caused by the tilt-effect shows more similarities with an absorber, the changes in its dSCDs might however be larger. This depends on the fitting interval, spectral resolution and the respective absorber and cannot be answered in general.**

12. P19, section Conclusions It would further round up the conclusions section if the authors conclusion about the impact of the tilt-effect on the accuracy of dSCD of trace gases would be added.

    **As stated in the previous answer, a general answer on the effect of the tilt-effect on the retrieved dSCDs cannot be given in general and depends on various parameters. We added this to the conclusions.**

subsectionTechnical corrections

1. P3, L23 ... around each of the lines (dashed, cyan and red) ... (possibly obsolete, see comment 2. in former section)

**see above**

2. P8, Figure 2 When I am not mistaken, the bottom figure is never mentioned (or referred to) in the text explicitly. I suggest rectify this or to remove the bottom figure.

   **We added a reference to the bottom figure of Figure 2 to the paragraph about the connection between tilt-effect and colour-index (2.4).**

3. P9, L10 Instrument slit function H should be lowercase h as introduced earlier in the text.

   **fixed.**

4. P9, L20 Table 1 is not mentioned yet in the text. This could be a place to do this.

   **done.**

5. P9, L29 Eq (17), LaTeX typesetting issue

   **fixed.**

6. P11, Table 2 caption Last sentence: squeeze definition

   **fixed.**

7. P17, L27 To these Gaussian instrument functions the derivation ...

   **fixed.**

8. P18, L28 ..., can result in a

   **fixed.**

9. P19, L2 ... artificial shifts and enhanced residuals ...

   **fixed.**

10. P19, L24 ... approximated from a given difference ...

    **fixed.**

**References**

[Chance et al., 2005] Chance, K., Kurosu, T. P., and Sioris, C. E. (2005). Under-sampling correction for array detector-based satellite spectrometers. *Appl. Opt.*, 44(7):1296–1304.

[Chance and Kurucz, 2010] Chance, K. and Kurucz, R. (2010). An improved high-resolution solar reference spectrum for earth's atmosphere measurements in the ultraviolet, visible, and near infrared. *Journal of Quantitative Spectroscopy and Radiative Transfer*, 111(9):1289 – 1295. Special Issue Dedicated to Laurence S. Rothman on the Occasion of his 70th Birthday.

[Haley et al., 2004] Haley, C. S., Brohede, S., Sioris, C., Griffioen, E., Murtaugh, D., McDade, I., Eriksson, P., Llewellyn, E., Bazureau, A., and Goutail, F. (2004). Retrieval of stratospheric $O_3$ and $NO_2$ profiles from odin optical spectrograph and infrared imager system (osiris) limb-scattered sunlight measurements. *Journal of Geophys. Res.*, 109:D16303.

[Peters et al., 2016] Peters, E., Pinardi, G., Seyler, A., Richter, A., Wittrock, F., Bösch, T., Burrows, J. P., Van Roozendael, M., Hendrick, F., Drosoglou, T., Bais, A. F., Kanaya, Y., Zhao, X., Strong, K., Lampel, J., Volkamer, R., Koenig, T., Ortega, I., Piters, A., Puentedura, O., Navarro, M., Gómez, L., Yela González, M., Remmers, J., Wang, Y., Wagner, T., Wang, S., Saiz-Lopez, A., García-Nieto, D., Cuevas, C. A., Benavent, N., Querel, R., Johnston, P., Postylyakov, O., Borovski, A., Elokhov, A., Bruchkouski, I., Liu, C., Hong, Q., Liu, H., Rivera, C., Grutter, M., Stremme, W., Khokhar, M. F., and Khayyam, J. (2016). Investigating differences in doas retrieval codes using mad-cat campaign data. *Atmospheric Measurement Techniques Discussions*, 2016:1–31.

[Sioris et al., 2006] Sioris, C. E., Kovalenko, L. J., McLinden, C. A., Salawitch, R. J., Van Roozendael, M., Goutail, F., Dorf, M., Pfeilsticker, K., Chance, K., von Savigny, C., Liu, X., Kurosu, T. P., Pommereau, J.-P., Bösch, H., and Frerick, J. (2006). Latitudinal and vertical distribution of bromine monoxide in the lower stratosphere from scanning imaging absorption spectrometer for atmospheric chartography limb scattering measurements. *Journal of Geophysical Research: Atmospheres*, 111(D14). D14301.

[Slijkhuis et al., 1999] Slijkhuis, S., von Bargen, A., Thomas, W., and Chance, K. (1999). Calculation of undersampling correction spectra for doas spectral fitting.

[Wang et al., 2017] Wang, Y., Beirle, S., Hendrick, F., Hilboll, A., Jin, J., Kyuberis, A. A., Lampel, J., Li, A., Luo, Y., Lodi, L., Ma, J., Navarro, M., Ortega, I., Peters, E., Polyansky, O. L., Remmers, J., Richter, A., Rodriguez, O. P., Roozendael, M. V., Seyler, A., Tennyson, J., Volkamer, R., Xie, P., Zobov, N. F., and Wagner, T. (2017). Max-doas measurements of hono slant column densities during the mad-cat campaign: inter-comparison and sensitivity studies on spectral analysis settings. *Atmospheric Measurement Techniques Discussions*, 2017:1–38.